# Staged Strategies to Deal with Complex, Giant, Multi-Fossa Skull Base Tumors

**DOI:** 10.3390/brainsci13060916

**Published:** 2023-06-06

**Authors:** Brandon Edelbach, Miguel Angel Lopez-Gonzalez

**Affiliations:** 1School of Medicine, Loma Linda University, Loma Linda, CA 92354, USA; bedelbach@students.llu.edu; 2Department of Neurosurgery, Loma Linda University Medical Center East Campus, Loma Linda, CA 92354, USA

**Keywords:** meningioma, pituitary adenoma, endonasal endoscopic transsphenoidal approach, anterior clinoidectomy, transcavernous

## Abstract

Given the complex and multifaceted nature of resecting giant tumors in the anterior, middle, and, to a lesser extent, the posterior fossa, we present two example strategies for navigating the intricacies of such tumors. The foundational premise of these two approaches is based on a two-stage method that aims to improve the visualization and excision of the tumor. In the first case, we utilized a combined endoscopic endonasal approach and a staged modified pterional, pretemporal, with extradural clinoidectomy, and transcavernous approach to successfully remove a giant pituitary adenoma. In the second case, we performed a modified right-sided pterional approach with pretemporal access and extradural clinoidectomy. This was followed by a transcortical, transventricular approach to excise a giant anterior clinoid meningioma. These cases demonstrate the importance of performing staged operations to address the challenges posed by these giant tumors.

## 1. Introduction

Surgical excision of giant tumors in the anterior, middle, and partially in posterior fossa presents unique challenges due to the extension of the lesions in the sagittal, coronal, and axial planes. The extension of the lesions varies, and the surgical approach is tailored accordingly based on the size of the tumor. Several strategies have evolved for excising giant tumors in this environment. These multiple strategies include pterional craniotomy, modified pterional craniotomy, the cranio-orbitozygomatic approach, middle fossa craniotomy through anterior transpetrosal and posterior transpetrosal approaches, as well as endoscopic endonasal and expanded approaches [1,2,3,4]. Furthermore, depending on the characteristics of the tumor, a combination of these approaches may be used to ensure a thorough resection of the tumor. Giant pituitary adenomas and anterior clinoid meningiomas are two types of tumors that often require complex surgical strategies for complete resection due to their unique anatomical characteristics and potential complications [5,6,7,8].

Pituitary adenomas are intracranial tumors that account for 5–14% of surgically resected lesions [5]. Common symptoms of these tumors include bitemporal hemianopsia, headaches, and endocrine dysfunction [6]. Pituitary adenomas are graded according to the extent of invasion of local anatomical structures. Grade I represents pituitary adenomas that are limited to the sellar region. Grade II represents invasion into the cavernous sinus. Grade III is characterized by the elevation of the dura of the superior wall of the cavernous sinus. Supradiaphragmatic–subarachnoid extensions are characteristic of Grade IV pituitary adenomas [9]. Pituitary adenomas can exhibit invasive extensions that follows anatomic pathways through or around dura of the sellar region, creating diverse tumor morphology [10,11]. The structural diversity of these tumors, combined with the intricate anatomical structures present in the anterior, middle, and posterior fossa, often makes the removal of giant pituitary adenomas a complex task [12]. The removal of pituitary adenomas may be associated with postoperative complications such as cerebrospinal fluid (CSF) leak, diabetes insipidus, additional pituitary dysfunction, visual deterioration, and hydrocephalus [13], while a subtotal resection can be associated with pituitary apoplexy in up to 5.65% of cases [14]. Mitigation of these symptoms has been attempted through the use of several different surgical strategies with the intent of reducing the morbidity of this operation. The microscopic transsphenoidal approach has historically been associated with lower morbidity compared to transcranial strategies. However, with the development of endoscope-assisted microneurosurgery, surgical risks have been further reduced. As a result, the use of the endoscopic endonasal transsphenoidal (EET) approach has flourished in recent decades [15].

Contemporary methods of EET surgery involve the resection of the middle turbinate, with or without dissection of a nasoseptal flap, resection of the posterior septum, and exposure of the sphenoid ostium [16,17]. Often, the initial stage is completed by the otorhinolaryngology surgery team, and the subsequent steps are completed in collaboration with the neurosurgery team [18]. The sphenoid ostium is opened by removing the anterior wall of the sphenoid sinus, which allows access to the sella turcica [19]. Further extension may be necessary, and the choice of approach (transplanum, transclival, pterygomaxillary, or transorbital) will depend on the morphology of the pituitary adenoma to ensure proper visualization of the tumor and infundibular region.

Meningiomas originate from arachnoid cells and are classified into three WHO grades based on their histopathology [20,21]. Of significance, Grade I meningiomas have an increased likelihood of developing at the skull base [22]. The symptoms associated with meningiomas are often non-specific (headache, seizure, cognitive change, vertigo, ataxia), but may involve cranial nerve deficits dependent upon tumor morphological distribution. When it comes to skull base tumors, cranial nerve deficits are more likely to occur [7,23]. Furthermore, anterior clinoid meningiomas are known to have the propensity to present with visual impairments and exophthalmos [24]. Due to their slow growth rate and tendency to present later in life, small meningiomas (less than 3 cm) are often left untreated and followed with serial imaging. However, larger meningiomas, particularly those causing symptoms, are typically surgically resected and treated with stereotactic radiation, based on the WHO grading scale [8].

Similar to pituitary adenomas, giant anterior clinoid meningiomas present unique challenges for complete resection. A significant challenge arises from the tumor’s tendency to compress the neurovascular structures associated with the anterior clinoid process [25]. The extent of tumor invasion also plays a significant role in the challenges associated with completely excising giant anterior clinoid meningiomas. The Al-Mefty classification system is based on the microanatomy of the tumor and is divided into three groups. Group I clinoid meningiomas extend over the inferior aspect of the anterior clinoid process and encircle the internal carotid artery; group II clinoid meningiomas are derived from the superior portion of the anterior clinoid process and are covered by arachnoid; group III clinoid meningiomas are derived from the optic foramen [26]. The typical method used for resecting giant anterior clinoid meningiomas consists of a pterional transsylvian approach, although alternative methods including orbitocranial or cranioorbitozygomatic approaches combining both intra- and extradural techniques have been proposed for giant anterior clinoid meningiomas [27,28,29]. However, the utilization of these techniques varies and is primarily dependent on the unique characteristics of the tumor and the neurosurgeon’s level of expertise.

## 2. Results

### 2.1. Case 1

A 65-year-old female presented with encephalopathy secondary to a suprasellar mass with symptoms of chronic bitemporal hemianopsia, which had worsened over the course of three years. Prior to admission, she experienced a sudden deficit and could only perceive light and movement. Due to a deterioration in vision, urgent surgical intervention was recommended. A CT scan with contrast and navigation sequences revealed a heterogeneously enhancing mass measuring 6 × 2.4 × 6 cm that originated from the sella and extended inferiorly into the sphenoid sinus and anterior fossa. Additionally, there was a superior extension into the left corona radiata, clivus, and left middle fossa. The mass also encased the left internal carotid artery and extended to the bilateral cavernous sinus and also to the left crural, ambiens, and cerebellopontine cistern. A 3 mm rightward midline shift in the left lateral ventricle was noted due to mass effects, as shown in Figure 1 and Figure 2. Given the recent visual decline, preoperative MRI was unable to be obtained and the patient was taken urgently for surgery.

A staged procedure was planned due to the tumor’s lateral extension. Stress doses of steroids were given perioperatively and tapered to a maintenance dose. The patient continued with levothyroxine supplementation. The patient was taken to the operating room for stage 1 endoscopic endonasal tumor resection with the goal of resecting the intrasellar and suprasellar compartments of the tumor to alleviate pressure of the optic apparatus (Figure 3). The sellar component of the tumor was removed using skull base ring curettes, a side-cutting aspirator, and an ultrasonic aspirator. In the superior regions of the sellar component, patties were used to retract the arachnoid tissue, allowing for debulking of the tumor around the cavernous carotid arteries using endoscopic and microsurgical techniques. The midline segment of the tumor was removed, except the segments in the middle and posterior fossa due to the angle of approach, as shown in Figure 3.

The patient was brought back into the operating room for stage 2 a week later, which involved a left-sided modified pterional transcavernous and transsylvian approach to remove the tumor from the middle and posterior fossa. The procedure required careful dissection around the left supraclinoid carotid, left middle cerebral artery, left posterior communicant artery, bilateral posterior cerebral arteries, and basilar apex. The oculomotor nerve was fully decompressed from a cavernous sinus tumor, and a small segment of fibrous tumor was left attached to the left cavernous carotid artery. The surgical site was closed using a dura substitute, surgical glue, and fat harvested from the abdomen. Postsurgical imaging of subtotal resection is shown in Figure 4 and Figure 5, and an illustrated summary of the staged approach is presented in Figure 6. At the 6-month follow up, the patient maintained light and movement perception, and the postoperative left oculomotor palsy had improved 6 months after surgery. No further deficits were encountered.

### 2.2. Case 2

A 74-year-old male presented with cognitive decline over several months, as well as memory and visual deficits, accompanied by a significant decline in balance. Imaging revealed a large extra-axial mass, measuring 4.3 × 6.3 cm, located at the right anterior clinoid process. The mass showed significant suprasellar and cavernous sinus extensions into the bilateral anterior fossa, middle fossa, and partially within the posterior fossa in retroclinoid space, overall resulting in significant compression of the optic chiasm apparatus and brainstem. There was significant vasogenic edema and obstructive hydrocephalus. This is illustrated in Figure 7.

The patient was brought into the operating room, and it was planned to resect the tumor in two stages during the same operation if it showed hard consistency on the first approach. During craniotomy stage one, a modified right-sided pterional, transzygomatic, and pretemporal approach was utilized, which included extradural clinoidectomy and optic canal decompression. This was followed by transsylvian dissection. The dissection was continued to the superior aspect of the cavernous sinus, allowing visualization of the anterior cerebral arteries and optic nerves. The optic canal was decompressed, and dissection was continued toward the internal carotid arteries and middle fossa, from which a portion of the tumor was resected. The tumor was then meticulously separated from the right internal carotid artery, the right middle cerebral artery, the lateral lenticulostriate branches, the bilateral anterior cerebral arteries (A1 and A2 segments), the right anterior choroidal artery, and the right posterior communicant artery. Then, the approach continued with tumor devascularization, which was achieved by capsule electrocoagulation and by placing aneurysm clips on two main arteries that supplied the tumor. Neurophysiology monitoring was used to provide standard assistance during the procedure and confirm stable somatosensory evoked potentials and electroencephalography during temporary clipping of the vasculature.

As the entire tumor showed a hard consistency, the superior aspect of the lesion was inaccessible through this approach. Therefore, stage two was initiated with a separate right frontal craniotomy using the same incision. A transfrontal, transcortical, and transventricular microsurgical approach was performed on the right side. The giant anterior clinoid meningioma tumor was visualized through this approach and successfully debulked, exposing the anterior cerebral arteries, right internal carotid artery, and middle cerebral artery. The two large arterial tumor feeders with the temporary clips were then electrocoagulated and permanent titanium clips applied. After the tumor was sufficiently removed, fat was harvested from the abdomen and placed in the pterional area. Additionally, a right frontal external ventricular drain was left in place under direct visualization. Postoperative imaging is illustrated in Figure 8, while a summary of the staged approach is shown in Figure 9. External ventricular drain was removed on postoperative day 5. At 6 months after surgery, there was significant improvement in ambulation and cognition without additional neurological deficits.

Pathologic examination of the tumor also confirmed a low-grade meningioma. The Ki67 indices were low, with mild focal elevation. E-cadherin, BAP-1, and PR stains were positive and GFAP stains indicated glial tissue along edges without any indication of brain invasion.

## 3. Discussion

### 3.1. Giant Pituitary Macroadenoma Surgery

In 1992, Jankowski et al. [30] described the first successful endonasal endoscopic resection of a pituitary adenoma in three patients. This represented a major transition from the previously popular method of microscopic transsphenoidal resection via a sublabial or endonasal approach. The endoscopic endonasal approach has been associated with significant improvements in morbidity and mortality associated with the removal of pituitary adenomas [31].

The advancements in the resection of pituitary adenoma using EET resection have been well documented [32,33]. EET surgery has been reported to achieve resection rates greater than 80% in tumors with a volume of 18 cm^3^, as well as gross total resection rates up to 44% [34]. Postoperatively, there was a significant improvement in visual function (82%) and pituitary function (20–72%) in those who presented with pituitary dysfunction [33,34,35]. McLaughlin et al. concluded that the use of endoscopy allowed for the removal of adenomas in an additional 36% of patients, thanks to improved visualization. Furthermore, the use of endoscopy was accentuated in patients presenting with tumors larger than 2 cm, permitting the removal of 54% of pituitary adenomas [32].

However, significant complications have been reported with this procedure. As many as 37% of patients have experienced complications, which include sinusitis (13.7%), CSF leak (9.6–11.4%), and SIADH (4.1%), as well as headache, epistaxis, meningitis, and hydrocephalus in a minority of patients [33,35]. Additional clinical reports have detailed complications of diabetes insipidus at rates as high as 25–45.5%, with a minority experiencing ischemic stroke [34,35].

It is relevant to mention that the endoscopic endonasal approach, with an expanded transtubercular approach, can still be associated with a lower degree of resection when the tumor has a significant lateral extension, harder consistency, or cavernous sinus extension [36], and an open transcranial approach has a significant role in its surgical treatment [37].

### 3.2. Surgical Pearls

Giant pituitary macroadenomas present a significant challenge due to their extension into various anatomical compartments. In our example case, the tumor was extended into three different anatomical compartments of the skull base involving the neurovascular structures. A meticulous review of all available images is necessary to plan the different steps of the operation. After obtaining appropriate medical clearance and ensuring stability, surgery should be performed promptly. With a significant visual deficit, the initial planned step was to debulk the tumor mass inferiorly through an endoscopic endonasal approach with a transplanum sphenoidale extension to alleviate ventral compression of the optic chiasm. A combination of microsurgical and endoscopic tumor dissection techniques was required. Usually, as in this case, the capsule of a pituitary adenoma is harder in consistency, requiring careful dissection from the anterior cerebral arteries, and it is of utmost importance to maintain the anatomical landmarks respecting the trajectory of the optic apparatus. Hypervascularity is a common feature of tumors with a hard consistency. To address this, a combination of different endonasal bipolar electrocoagulators may be necessary, including long and fine-tipped regular bipolar ones. The significant lateral, superior, and posterior extension of the tumor was the deciding factor for a staged operation, where a modified pterional, extradural anterior clinoidectomy, transcavernous, and transsylvian approach allowed access to all of these compartments. After opening the intradural space, an important goal is to establish a plan for resecting a giant tumor in sectors: (a) there were multiple critical anatomical landmarks for the anterior sector of the tumor including optic nerves and chiasm, anterior cerebral arteries, and anterior communicant artery region complex; (b) the central sector with suprasellar and intercarotid space, dealing with bilateral internal carotid arteries, left posterior communicant and choroidal arteries identifying normal vasculature from feeding tumor vessels; (c) posterior sector with tumor extension to interpeduncular fossa and left cerebellopontine angle cistern, dealing with bilateral posterior cerebral arteries (P1 segments), basilar apex, and thalamo-perforating vessels. If a tumor segment exhibits tight adhesions to neurovascular structures, making it difficult to identify a clear cleavage plane, we recommend avoiding the risk of neurovascular injury, requiring leaving some tumoral tissue in place. As part of our routine, we always keep aneurysm clips, clip appliers, and cerebral bypass instruments available in the operating room in case they are needed. During the dissection of the lateral wall of the cavernous sinus, it is important to identify the normal trajectory of the nerves, especially the oculomotor and trochlear nerves, before accessing either the roof or lateral wall of the cavernous sinus. The oculomotor nerve is highly sensitive to manipulation and requires adequate release from the oculomotor cistern and lateral wall of the cavernous sinus after peeling off the dura from the middle fossa. A temporary oculomotor deficit often improves within weeks or months after surgery. This possibility should be thoroughly discussed with patients and their families prior to the operation. The consistency and vascularity of the tumor within the cavernous sinus will determine the extent of resection required. Postoperative care in the intensive care unit, along with an adequate protocol for managing diabetes insipidus and individualized hormone replacement, is essential.

### 3.3. Giant Anterior Clinoid Meningioma Surgery

There is significant debate regarding the most effective strategy for removing giant anterior clinoid meningiomas, mainly due to the unique challenges discussed earlier in resecting these tumors [38,39,40,41,42]. Two popular techniques involve either a vascular or skull base perspective. The vascular strategy of tumor debulking involves dissecting the sylvian fissure to trace the middle cerebral artery to the internal carotid artery while removing the tumor and its associated perforating arterial supply [43]. However, this school of thought is commonly criticized due to the strain placed on the sylvian fissure [26]. The alternative solution to this issue involves performing a pretemporal dissection and an anterior clinoidectomy to expand the operating field while minimizing brain retraction at the sylvian fissure [44]. This technique is associated with several advantages, such as early optic nerve decompression, early identification of the internal carotid artery, and associated devascularization of the meningioma [44]. This is associated with a decreased rate of complications. The occurrence of postoperative vascular complications has been reported in 2% of cases; cranial nerve deficits occurred in 5.5% of patients; and the overall patient mortality rate was 1.2% [45].

Further modifications to the anterior clinoidectomy have been reported. This includes extradural, intradural, and hybrid approaches to this technique. However, the merits of each of these subtechniques are still debated [45]. The use of preoperative embolization has been reported, but it is also problematic. External carotid artery branches are safer to embolize, with limited opportunities for branches arising directly from the internal carotid artery. One clinical trial found a complication rate of 12% associated with this practice, which seemed to provide little benefit to the quality of patient postsurgical recovery [46]. Additional methods are still under exploration. One such method is the Dolenc approach, which involves an extradural clinoidectomy and transdural debulking of the tumor. A study reported that 67% of patients had better vision outcome [38,39]. The gross total resection rate was 30.4%, and partial resections were achieved in 34.8% of surgeries [38,41].

When considered as a complete entity, giant anterior clinoid meningiomas had a gross total resection rate of 64.2% [45], while 25% of cases resulted in subtotal resections [41]. The reported operative mortality was 6.7%, and recurrence was observed in 11.8% of cases [40]. According to Nanda et al., four out of thirty-six patients who underwent surgery for clinoidal meningiomas experienced recurrence, with a median duration of 89 months, and one patient required repeat surgery [41]. Furthermore, it is worth noting that gross total resections of group I giant anterior clinoid meningiomas were limited to only 11.8% due to the anatomical difficulties associated with this type of tumor [45]. The minimally invasive options such as the endoscopic endonasal transtubercular approach or endoscopic-assisted supraorbital key-hole approach are ideal for midline lesions such as tuberculum sella meningiomas [46], although those options have a limited role in anterior clinoid meningioma and even less so in a giant tumor given its anatomical skull base implications [42,47].

### 3.4. Surgical Pearls

Meningiomas located on the anterior clinoid process may vary in size. A comprehensive surgical extension involving the anterior, middle, posterior fossa, and intraventricular areas requires a detailed analysis of preoperative imaging for effective planning. This case example involved a lesion larger than 6 cm originating from the right anterior clinoid process with a significant suprasellar and lateral cavernous sinus extension into the bilateral anterior fossa, middle fossa, and a segment of the posterior fossa, with significant compression of the optic chiasm and brainstem. A preoperative angiogram is routinely obtained for meningiomas located in this area to assess the possibility of embolizing branches from the external carotid artery. This is because branches of the internal carotid artery pose a higher risk for ischemic complications. The angiogram is obtained to define the blood supply to the tumor, regardless of whether it is possible to embolize it or not. This can provide information on the degree of displacement of the normal vasculature, collateralization, the presence of posterior communicant arteries, and cross-flow through anterior communicant arteries. All of these factors, combined with preoperative magnetic resonance imaging and computer tomography, can help define the surgical strategy. Additional tools, such as neuronavigation and neuromonitoring with techniques including somatosensory evoked potentials, brainstem auditory evoked responses, and electroencephalography, are always helpful. Given the location of the tumor, vascular proximal control measures need to be planned, either cervical carotid with prep of the cervical area, petrous carotid through the middle fossa, or clinoid carotid. Occasionally, there is the need to perform temporary clipping of some feeding tumor vessels, and neuromonitoring provides feedback on physiological stability during these episodes. The type of craniotomy, such as pterional, modified pterional, orbitocranial, or cranio-orbito-zygomatic, can be selected based on the patient’s unique anatomy. Ideally, transzygomatic or cranio-orbito-zygomatic approaches can be used for tumors with significant superior extension, as in the case example presented. In addition, techniques such as pretemporal dissection and extradural clinoidectomy are recommended to partially devascularize and remove the origin of the tumor with early optic canal decompression, which can subsequently be released intradurally after opening the falciform ligament. After meticulous extradural dissection and careful attention to hemostasis, the initial intradural approach aims to explore the anatomical distortion caused by the tumor and to identify normal anatomy. This involves searching for optic nerves, oculomotor nerves, internal carotid arteries, and anterior and middle cerebral arteries. This tumor was wrapped around the right internal carotid artery and right middle cerebral artery. The goal was to divide it into sectors, starting with the lateral component in the middle fossa and around the middle cerebral artery. Central debulking was performed using high microsurgical magnification and an ultrasonic aspirator. Releasing a tumor from vascular structures such as the middle cerebral artery and towards the carotid bifurcation can be performed using constant micro-Doppler and neuronavigation to map the vascular trajectory. Once an arterial trunk is found, it can be followed proximally to remove the majority of the tumor. In cases where a tumor is calcified around the supraclinoid carotid artery or middle cerebral artery, it may be necessary to leave a cuff of the tumor to avoid causing unnecessary injury. Through the middle fossa approach, the sector adjacent to the crural and ambiens cisterns can be carefully resected while dissecting the posterior communicant artery and perforating vessels. If a tumor extends significantly in the inferior direction to the cerebellopontine angle, an anterior petrosectomy may be necessary. However, it was not required in this case. The anterior segment of the tumor is gradually dissected, following the ipsilateral and contralateral A1 segments of the anterior cerebral artery. After performing devascularization, central and superior debulking of the mass was continued, although given the hard consistency of the tumor and its capsule, it was decided to perform a staged frontal craniotomy. This procedure was conducted through a small right frontal coronal craniotomy, with a small transcortical approach used to reach the right lateral ventricle. Central debulking was then performed, followed by medial dissection of the capsule through the arachnoid plane from the anterior cerebral artery A2 and A3 segments and lateral dissection from the superior trunk of the middle cerebral artery. Finally, the dissection was carefully performed from the superior aspects of optic nerves and chiasm.

## 4. Conclusions

Giant tumors located in the skull base pose significant challenges due to their size, location, and proximity to critical neurovascular structures. Despite these challenges, current advances in surgical techniques and new technology have made it possible to safely remove giant skull base tumors. Neurosurgeons may employ a combination of open and minimally invasive approaches, such as endoscopic techniques, microvascular dissection, and microanastomosis, if necessary. The success of skull base surgery for giant tumors depends on several factors, including the tumor’s location and size, the patient’s overall health, the appropriate selection of treatment, and the expertise of the surgical team. Surgery for these patients is preferably performed at highly specialized centers with a multidisciplinary approach in order to provide a higher chance of success and long-term survival.

## Figures and Tables

**Figure 1 brainsci-13-00916-f001:**
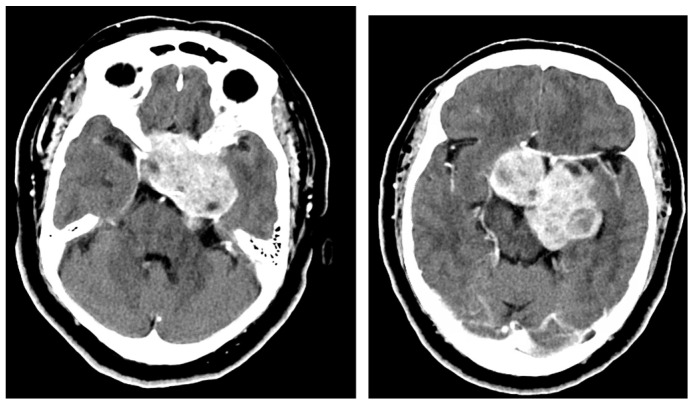
Case 1 giant pituitary adenoma prior to stage 1 resection. CT with contrast shows extension from anterior cranial fossa anteriorly at planum sphenoidale, evident middle fossa involvement, left cavernous sinus and cerebellopontine angle.

**Figure 2 brainsci-13-00916-f002:**
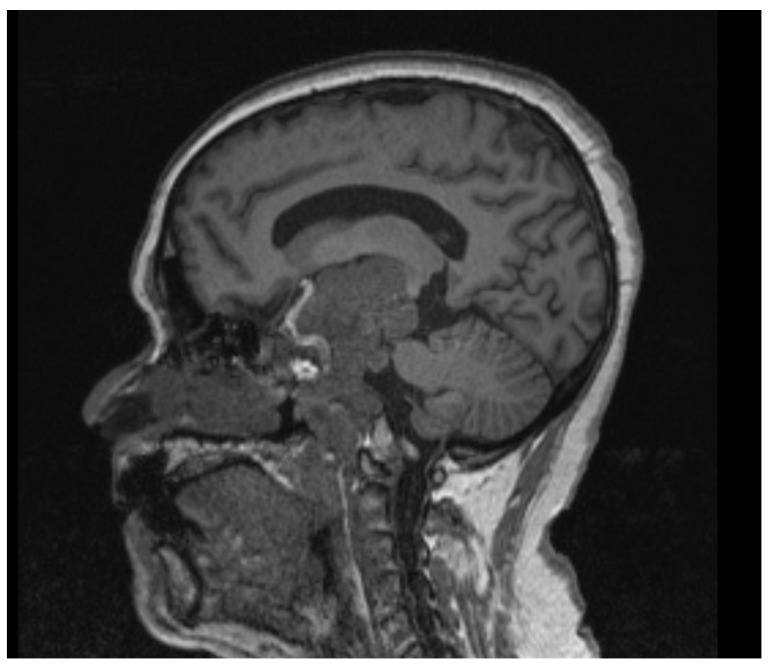
Case 1 giant pituitary adenoma sagittal MRI scan wo contrast.

**Figure 3 brainsci-13-00916-f003:**
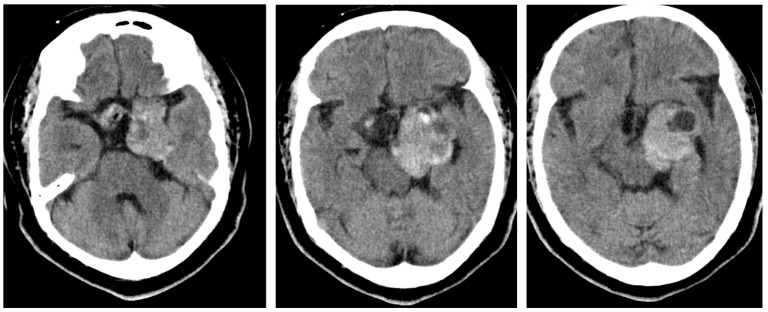
Poststage 1 resection CT. There are postsurgical changes related to the transsphenoidal approach debulking/partial resection of a large sellar/suprasellar mass with fat packing at the sellar floor. A large, residual, multilobulated, heterogeneously enhancing, partially cystic/partially solid mass measures 4.4 × 3.7 × 4.3 cm. The mass extends superiorly from the sella, exerting a mass effect superiorly and displacing the left basal ganglia/thalamus and laterally displacing the left temporal lobe/temporal horn. The mass is again seen within the left cavernous sinus. The residual mass extends posteriorly into the left cerebellopontine angle and perimesencephalic cistern, involves the left Meckel’s cave, and exerts mass effect on the left side of midbrain. The residual mass also extends to the left orbital apex, with apparent encasement of the intracranial portion of the left optic nerve and poor delineation of the optic chiasm.

**Figure 4 brainsci-13-00916-f004:**
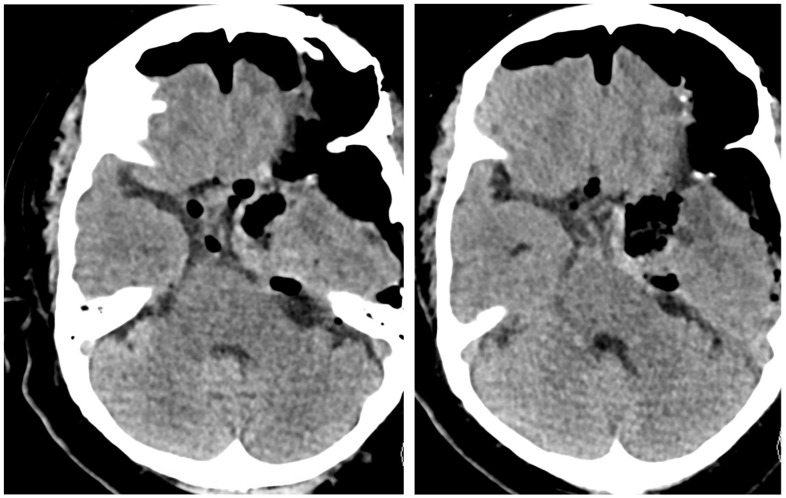
There is demonstration of postsurgical changes after a two-staged operation for subtotal resection of a giant pituitary macroadenoma.

**Figure 5 brainsci-13-00916-f005:**
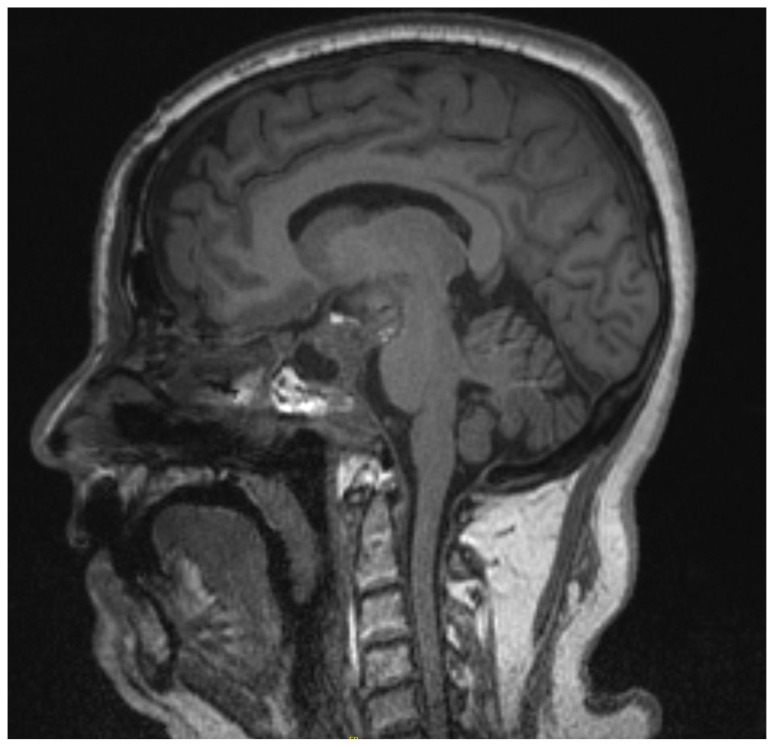
Poststage 2 resection. MRI. Postsurgical changes related to staged transphenoidal approach and subsequent left-sided transcranial skull base approach subtotal resection of a giant invasive pituitary macroadenoma, with fat grafting and fluid/hemorrhage within the surgical bed, with associated mass effect on the left side of the brainstem. Small residual tumor centered in the left cavernous sinus/Meckel’s cave with partial encasement of the left internal carotid artery.

**Figure 6 brainsci-13-00916-f006:**
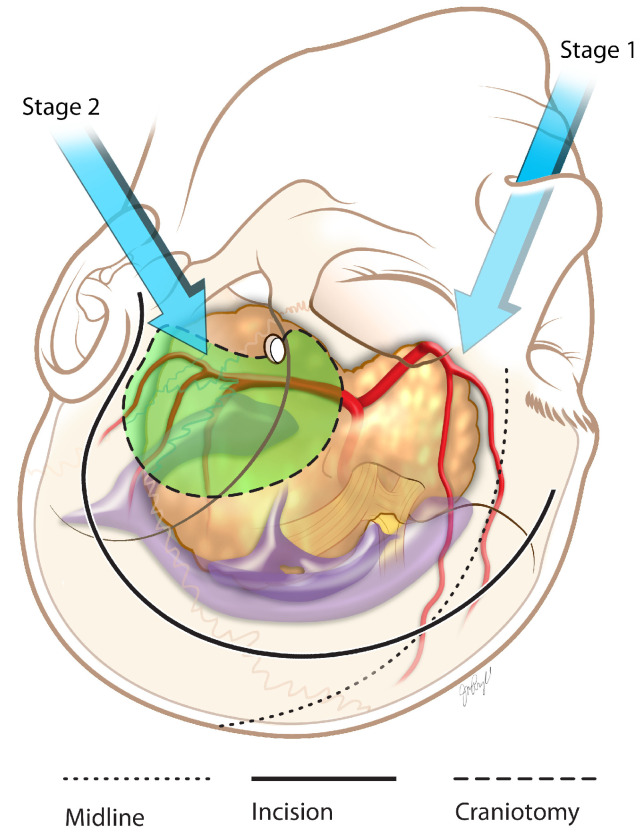
Giant pituitary adenoma two-staged approach illustration.

**Figure 7 brainsci-13-00916-f007:**
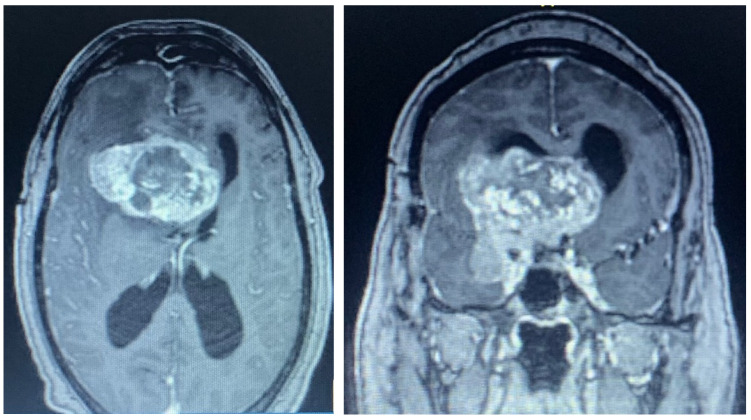
Preoperative axial (**left**) and coronal (**right**) MRI with contrast demonstrating a large, heterogeneously enhancing mass centered on the right anterior clinoid with cavernous sinus, suprasellar, middle fossa, anterior fossa, and intraventricular extension. It measures approximately 4.3 × 6.3 cm.

**Figure 8 brainsci-13-00916-f008:**
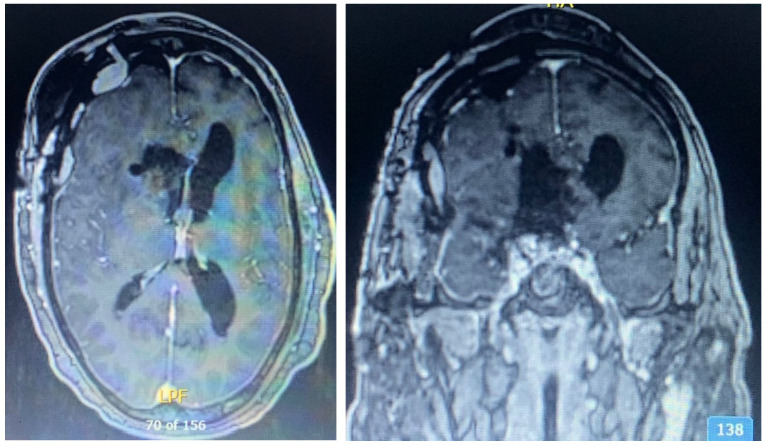
Postoperative axial (**left**) and coronal (**right**) MRI with contrast. Postsurgical changes related to right modified pterional, pretemporal approach and staged right frontal transcortical transventricular approach for resection of a large meningioma centered in the right anterior clinoid region.

**Figure 9 brainsci-13-00916-f009:**
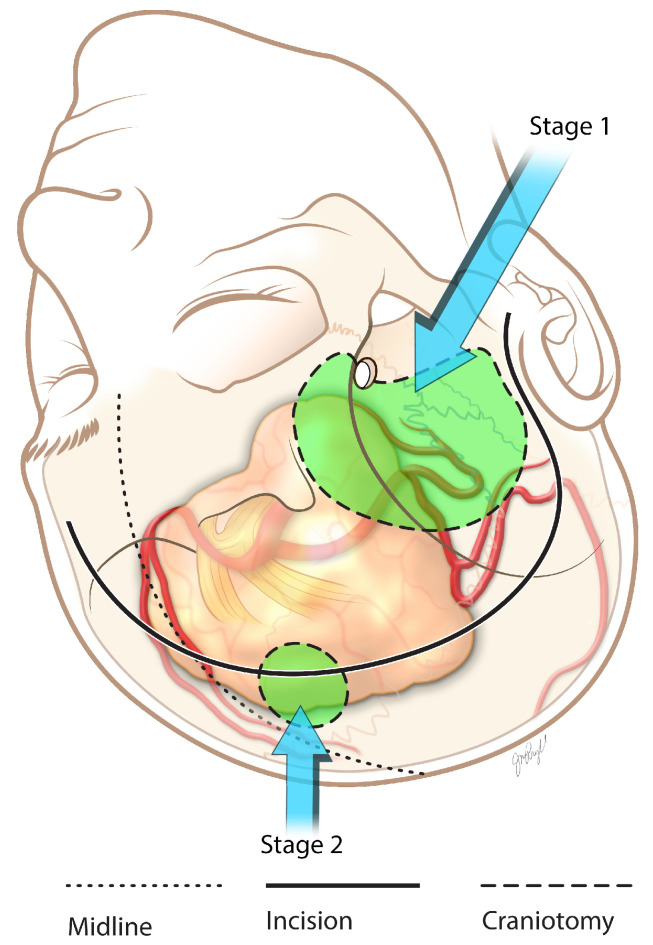
Giant anterior clinoid meningioma staging.

## Data Availability

Data sharing not applicable. No new data were created or analyzed in this study. Data sharing is not applicable to this article.

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
