# Peer review of "Staged Strategies to Deal with Complex, Giant, Multi-Fossa Skull Base Tumors"

_brainsci, 2023, doi:10.3390/brainsci13060916_

Round 1

Reviewer 1 Report

Well structured, well crafted desciption of systematic surgical approaches to complex intracranial skull base lesion in n=2 patients. I´ve read this report with interest. It provides good overview and thoughtful strategy on how to treat such challenging lesions. However, the methods described are not quite unique or novel, but the reader can clearly assess the message.

Just two minor points: In case #1: Any special endocrinological considerations peri-/postoperatively? Why did the authors not present the MRI, but the CT scan?

Author Response

Reviewer 1

Well structured, well crafted description of systematic surgical approaches to complex intracranial skull base lesion in n=2 patients. I´ve read this report with interest. It provides good overview and thoughtful strategy on how to treat such challenging lesions. However, the methods described are not quite unique or novel, but the reader can clearly assess the message.

Response: We really appreciate your above comments.

Just two minor points: In case #1: Any special endocrinological considerations peri-/postoperatively? Why did the authors not present the MRI, but the CT scan?

Response: Patient received stress dose of hydrocortisone perioperatively with subsequent taper postoperatively to reach maintenance dose. She was also receiving already levothyroxine supplementation. Patient presented to emergency department with the large mass and recent visual decline. We elected to bring her urgently to operating room for first endonasal stage surgery only with the referred preoperative CT scan with and without contrast (including navigation sequences).

Described on lines 102, and 108 to 113.

Reviewer 2 Report

The authors present two impressive surgical cases managing difficult pathological entities. but there is absolutely nothing new in this paper, even though the cases are presented nicely. surgeons have been combining these approaches for many years, and it's all been shown before. also, these cases are not truly post fossa lesions, so the authors really shouldn't be commenting on this entity within the context of the cases presented.

fine

Author Response

Reviewer 2

The authors present two impressive surgical cases managing difficult pathological entities. but there is absolutely nothing new in this paper, even though the cases are presented nicely. surgeons have been combining these approaches for many years, and it's all been shown before. also, these cases are not truly post fossa lesions, so the authors really shouldn't be commenting on this entity within the context of the cases presented.

Response: We really appreciate your comments. We agree that these types of surgeries are not a novel approach, although it can exemplify strategies to deal with these tumors. We fully agree that involvement on the upper segment of posterior fossa is minor, but still, being a small segment of tumor involving left crural, ambiens, and cerebellopontine cistern as in example one, and retroclinoid segment on case example two, merits the surgeon to analyze on details what approach or approaches to use in order to reach all involved anatomical compartments.

Modified title, abstract and introduction accordingly on lines 2,3, 10, 19, 24 and 25.

Reviewer 3 Report

The manuscript "Staged strategies to deal with giant anterior, middle, and posterior fossa tumors" by Edelbach and Lopez-Gonzalez presents two elaborately drafted reports of patients presenting with giant tumors of the skull base that posed a substantial surgical challenge to the team. Both patients necessitated a differentiated surgical approach and the authors render a fascinating and captivating report of their approach. The cases required skill and good planning which is highlighted by the multimodal and interdisciplinary approach.

This article addresses a complex and technically challenging issue in neurosurgery: the resection of giant tumors located at the skull base. Given the intricate anatomy of these regions, sharing successful strategies can help in enhancing patient outcomes and fostering safer surgical approaches embedded in real-world scenarios which is of significant practical value. By reporting real-world examples, the authors allow other clinicians to better understand the potential situations they might encounter and how they can be managed. The figures and illustrations are very helpful and insightful, elegantly rounding off the report.

minor comments:

* keywords shouldn't be numbered (or should they?)

* line 80: "of invade" can be removed, superfluous "the" towards the end as well

* line 105: insert a space between the quantity and unit ("3 mm")

* the discussion section 3.2 repeats too much of the description of the procedure (starting around line 240); this could be simplified

* line 368: superfluous "reach"

* line 377: fix "skull baser tumors"

Author Response

Reviewer 3

The manuscript "Staged strategies to deal with giant anterior, middle, and posterior fossa tumors" by Edelbach and Lopez-Gonzalez presents two elaborately drafted reports of patients presenting with giant tumors of the skull base that posed a substantial surgical challenge to the team. Both patients necessitated a differentiated surgical approach and the authors render a fascinating and captivating report of their approach. The cases required skill and good planning which is highlighted by the multimodal and interdisciplinary approach.

This article addresses a complex and technically challenging issue in neurosurgery: the resection of giant tumors located at the skull base. Given the intricate anatomy of these regions, sharing successful strategies can help in enhancing patient outcomes and fostering safer surgical approaches embedded in real-world scenarios which is of significant practical value. By reporting real-world examples, the authors allow other clinicians to better understand the potential situations they might encounter and how they can be managed. The figures and illustrations are very helpful and insightful, elegantly rounding off the report.

Response: We really appreciate your above comments.

Minor comments:

* keywords shouldn’t be numbered (or should they?)

Response: The editorial draft comes with this pattern. I will defer this to Editorial Office for this modification.

* line 80: “of invade” can be removed, superfluous “the” towards the end as well

Response: Thank you, completely agree, removed those.

* line 105: insert a space between the quantity and unit (“3 mm”)

Response: Thank you, added.

* the discussion section 3.2 repeats too much of the description of the procedure (starting around line 240); this could be simplified

Response: Agree with above comment. Removed lines 253 to 257.

* line 368: superfluous "reach"

Response: Agree and removed. Thank you.

* line 377: fix "skull baser tumors"

Response: Agree, fixed.

Reviewer 4 Report

The present paper about the staged management of giant anterior, middle and posterior cranial fossa lesions is well written and documented; on the other hand although the authors report 2 very challenging  and very well managed lesions the study does not add neither new information nor new management strategies to the exisiting body of literature. Further more the authors do not at all discuss the high risk of tumor apoplexy in statgeded procedures wich represent a crucial issue. Finally relevant literature is missing as follow: Chibbaro S, Signorelli F, Milani D, Cebula H, Scibilia A, Bozzi MT, Messina R, Zaed I, Todeschi J, Ollivier I, Mallereau CH, Dannhoff G, Romano A, Cammarota F, Servadei F, Pop R, Baloglu S, Lasio GB, Luca F, Goichot B, Proust F, Ganau M. Primary Endoscopic Endonasal Management of Giant Pituitary Adenomas: Outcome and Pitfalls from a Large Prospective Multicenter Experience. Cancers (Basel). 2021Jul 18;13(14):3603. doi: 10.3390/cancers13143603. PMID: 34298816; PMCID:

PMC8304085.

Author Response

Reviewer 4

The present paper about the staged management of giant anterior, middle and posterior cranial fossa lesions is well written and documented; on the other hand although the authors report 2 very challenging  and very well managed lesions the study does not add neither new information nor new management strategies to the exisiting body of literature. Further more the authors do not at all discuss the high risk of tumor apoplexy in statgeded procedures wich represent a crucial issue. Finally relevant literature is missing as follow: Chibbaro S, Signorelli F, Milani D, Cebula H, Scibilia A, Bozzi MT, Messina R, Zaed I, Todeschi J, Ollivier I, Mallereau CH, Dannhoff G, Romano A, Cammarota F, Servadei F, Pop R, Baloglu S, Lasio GB, Luca F, Goichot B, Proust F, Ganau M. Primary Endoscopic Endonasal Management of Giant Pituitary Adenomas: Outcome and Pitfalls from a Large Prospective Multicenter Experience. Cancers (Basel). 2021Jul 18;13(14):3603. doi: 10.3390/cancers13143603. PMID: 34298816; PMCID:

PMC8304085.

Response: We appreciate significantly your comments. We agree that such approaches are not novel, although are current examples of giant tumors requiring staged surgeries, with the intention to share to young skull base surgeons the relevance to keep in their armamentarium the open skull base surgery skills, and combining those with minimal invasive techniques, tailoring to the unique needs for each patient.

Appreciate your comment and concern regarding risk of pituitary apoplexy with subtotal resection which through a systematic review was found to be present in 5.65% of cases (added reference and mentioned on lines 51 and 52). In the case example presented, our plan was to perform the staged surgery during the same hospital admission. Additionally, as mentioned on the recommended reference, expanded endonasal approach alone can be associated with lesser degree of resection when tumor has a significant lateral extension, harder consistency, or cavernous sinus extension (lines 240 to 243, included the provided reference).

Reviewer 5 Report

The authors present a case series of 2 cases with (rare) giant tumors of the skull base and their 2 step approach to achieve a maximal radicality and good surgical outcome.

Finally, they discussed giant pituitary macro adenomas and clinoid meningiomas. 

The manuscript is well written. The cases are well presented. The authors could improve their manuscript according to the following aspects:

1. please add video of the different steps of the surgical procedure of these two cases

2. please add in the discussion following literature and the pros and cons of endonasal and transcranial approaches and minimally invasive surgeries

The extended endoscopic approach to perisellar and skull base lesions: is one nostril enough?

Oertel J, Senger S, Linsler S.Neurosurg Rev. 2020 Dec;43(6):1519-1529. doi: 10.1007/s10143-019-01171-8. Epub 2019 Sep 16.   Endoscopic Assisted Supraorbital Keyhole Approach or Endoscopic Endonasal Approach in Cases of Tuberculum Sellae Meningioma: Which Surgical Route Should Be Favored? Linsler S, Fischer G, Skliarenko V, Stadie A, Oertel J.World Neurosurg. 2017 Aug;104:601-611. doi: 10.1016/j.wneu.2017.05.023. Epub 2017 May 13.

Author Response

Reviewer 5

The authors present a case series of 2 cases with (rare) giant tumors of the skull base and their 2 step approach to achieve a maximal radicality and good surgical outcome.

Finally, they discussed giant pituitary macro adenomas and clinoid meningiomas.

The manuscript is well written. The cases are well presented. The authors could improve their manuscript according to the following aspects:

  1. please add video of the different steps of the surgical procedure of these two cases

Response: Agree that surgical video would have been a great addition, although no video was obtained.

  1. please add in the discussion following literature and the pros and cons of endonasal and transcranial approaches and minimally invasive surgeries

The extended endoscopic approach to perisellar and skull base lesions: is one nostril enough? Oertel J, Senger S, Linsler S. Neurosurg Rev. 2020 Dec;43(6):1519-1529. doi: 10.1007/s10143-019-01171-8. Epub 2019 Sep 16.

Endoscopic Assisted Supraorbital Keyhole Approach or Endoscopic Endonasal Approach in Cases of Tuberculum Sellae Meningioma: Which Surgical Route Should Be Favored? Linsler S, Fischer G, Skliarenko V, Stadie A, Oertel J. World Neurosurg. 2017 Aug;104:601-611. doi: 10.1016/j.wneu.2017.05.023. Epub 2017 May 13.

Response: Added related comment on line 240 to 243 regarding pituitary macroadenoma surgery, and for anterior clinoid meningioma from lines 325 to 330.

Round 2

Reviewer 2 Report

none

fine

Reviewer 4 Report

The authors have answered satisfactorily to the reviewers comments.